# β-Hydroxybutyrate, a Ketone Body, Potentiates the Antioxidant Defense via Thioredoxin 1 Upregulation in Cardiomyocytes

**DOI:** 10.3390/antiox10071153

**Published:** 2021-07-20

**Authors:** Shin-ichi Oka, Fan Tang, Adave Chin, Guersom Ralda, Xiaoyong Xu, Chengchen Hu, Zhi Yang, Maha Abdellatif, Junichi Sadoshima

**Affiliations:** 1Department of Cell Biology and Molecular Medicine, Rutgers New Jersey Medical School, Newark, NJ 07103, USA; okash@njms.rutgers.edu (S.-i.O.); ftang@atlanticspinecenter.com (F.T.); adelchin@gmail.com (A.C.); guersomr77@gmail.com (G.R.); glmfe1@hotmail.com (X.X.); ch965@gsbs.rutgers.edu (C.H.); yangz2@njms.rutgers.edu (Z.Y.); abdellma@njms.rutgers.edu (M.A.); 2Department of Cardiovascular Disease, Ningbo Medical Treatment Centre Li Huili Hospital, Ningbo 315000, China

**Keywords:** β-hydroxybutyrate (βHB), thioredoxin 1 (Trx1), cardiomyocytes

## Abstract

Thioredoxin 1 (Trx1) is a major antioxidant that acts adaptively to protect the heart during the development of diabetic cardiomyopathy. The molecular mechanism(s) responsible for regulating the Trx1 level and/or activity during diabetic cardiomyopathy is unknown. β-hydroxybutyrate (βHB), a major ketone body in mammals, acts as an alternative energy source in cardiomyocytes under stress, but it also appears to be involved in additional mechanisms that protect the heart against stress. βHB upregulated Trx1 in primary cultured cardiomyocytes in a dose- and a time-dependent manner and a ketogenic diet upregulated Trx1 in the heart. βHB protected cardiomyocytes against H_2_O_2_-induced death, an effect that was abolished in the presence of Trx1 knockdown. βHB also alleviated the H_2_O_2_-induced inhibition of mTOR and AMPK, known targets of Trx1, in a Trx1-dependent manner, suggesting that βHB potentiates Trx1 function. It has been shown that βHB is a natural inhibitor of HDAC1 and knockdown of HDAC1 upregulated Trx1 in cardiomyocytes, suggesting that βHB may upregulate Trx1 through HDAC inhibition. βHB induced Trx1 acetylation and inhibited Trx1 degradation, suggesting that βHB-induced inhibition of HDAC1 may stabilize Trx1 through protein acetylation. These results suggest that βHB potentiates the antioxidant defense in cardiomyocytes through the inhibition of HDAC1 and the increased acetylation and consequent stabilization of Trx1. Thus, modest upregulation of ketone bodies in diabetic hearts may protect the heart through the upregulation of Trx1.

## 1. Introduction

The prevalence of obesity and metabolic syndrome has increased tremendously in the past few decades [1]. Individuals with obesity and metabolic syndrome eventually develop insulin resistance and type II diabetes. More than half of diabetic patients develop cardiac dysfunction characterized by cardiac hypertrophy and diastolic dysfunction, collectively termed diabetic cardiomyopathy [2]. The hearts of patients suffering from obesity and metabolic syndrome often exhibit increased oxidative stress and mitochondrial dysfunction, the major driving force for the progression of diabetic cardiomyopathy [3].

Thioredoxin 1 (Trx1) is an evolutionarily conserved antioxidant that reduces oxidized proteins with disulfide bonds through thiol disulfide exchange reactions catalyzed at the catalytic center of Trx1 [4]. Trx1 also indirectly scavenges H_2_O_2_ through reduction of peroxiredoxins (Prdxs), Trx1-dependent peroxidases [5]. We have previously shown that mTOR and AMPK are the major direct targets of Trx1 in cardiomyocytes. Trx1 directly reduces oxidized cysteine residues in mTOR and AMPK through thiol disulfide exchange reactions, thereby maintaining their kinase activity even under stress conditions [6,7]. Although Trx1 becomes oxidized during thiol disulfide exchange reactions, it is reduced and recycled in the presence of thioredoxin reductase and NAPDH, an electron donor. We have recently shown that the increased production of NADPH through coordinated actions of Nampt (nicotinamide phosphoribosyltransferase), NAD kinase, and the pentose phosphatase pathway ameliorates high fat diet-induced diastolic dysfunction in mice, through the stimulation of Trx1 and glutathione [8]. Thus, Trx1 appears to be an important target for the treatment of diabetic cardiomyopathy, and it is therefore important to understand how endogenous Trx1 is regulated in diabetic cardiomyopathic hearts.

Ketone bodies are produced from fatty acids in the liver and act as an energy carrier to peripheral tissues when the glucose level is low during prolonged exercise or starvation or in the presence of low dietary carbohydrates [9]. Ketone bodies are also produced in the presence of insulin deficiency, such as Type I diabetes, when body cells cannot use blood glucose. Although excessive production of ketone bodies leads to life-threatening ketoacidosis in diabetic patients under glucose lowering treatment, increasing lines of evidence suggests that modest levels of ketone bodies, including β-hydroxybutyrate (βHB), play adaptive or salutary roles in the heart [9]. A ketogenic diet protects the heart against stress, such as ischemia and pressure overload [10,11]. Sodium glucose cotransporter 2 (SGLT2) treatment in diabetic patients increases circulating levels of βHB, which may play an important role in mediating some of the protective effects of SGLT2 inhibitors [12]. Ketone bodies serve as an alternative fuel source in a failing heart when fatty acid oxidation is reduced [13]. Importantly, ketone bodies facilitate protective effects not only through energy production, but also by affecting cellular signal transduction mechanisms [9]. For example, βHB inhibits class I histone deacetylases (HDACs) [14]. Ketone bodies also upregulate antioxidant enzymes such as catalase and superoxide dismutase [15,16]. However, the detailed molecular mechanisms through which βHB protects the heart independently of its role as an energy source remain unclear.

In this study we asked (1) whether βHB upregulates Trx1 in cardiomyocytes, and if so, (2) whether the salutary effect of βHB upon cardiomyocytes during oxidative stress is mediated through Trx1 and, (3) if so, how βHB leads to upregulation of Trx1 in cardiomyocytes.

## 2. Materials and Methods

### 2.1. Primary Cultures of Neonatal Cardiomyocytes and Fibroblasts

Primary cultures of cardiomyocytes were prepared from 1-day-old Crl: (WI) BR-Wistar rats. Cardiomyocyte- and fibroblast-rich fractions were obtained by centrifugation through a discontinuous Percoll gradient. Cardiomyocytes were cultured in complete medium containing Dulbecco’s Modified Eagle’s medium/F-12 supplemented with 5% horse serum, 4 μg/mL transferrin, 0.7 ng/mL sodium selenite, 2 g/L bovine serum albumin (fraction V), 3 mM pyruvate, 15 mM Hepes (pH 7.1), 100 μM ascorbate, 100 mg/L ampicillin, 5 mg/L linoleic acid, and 100 μM 5-bromo-2′-deoxyuridine. Culture dishes were coated with 0.3% gelatin. Fibroblasts were cultured with Dulbecco’s Modified Eagle’s medium with 10% fetal bovine serum. The cardiomyocyte viability was examined using trypan blue dye exclusion.

### 2.2. Immunoblot Analyses

Heart homogenates or cell lysates were prepared using a lysis buffer (50 mM Tris-HCl (pH 7.6), 1% Triton X-100, 10 mM EDTA, 150 mM NaCl, 50 mM NaF, 10 mM Sodium Butyrate, and protease inhibitor cocktail (Sigma-Aldrich, Saint Louis, MO, USA)). Total protein lysates (10–30 µg) were incubated with SDS sample buffer (Final concentration: 100 mM Tris (pH 6.8), 2% SDS, 5% glycerol, 2.5% 2-mercaptoethanol, and 0.05% bromophenol blue] at 95 °C for 5–20 min). For SDS-PAGE under non-reducing conditions, the lysates were prepared with lysis buffer containing 100 mM N-ethylmaleimide and SDS sample buffer without 2-mercaptoethanol. For immunoprecipitation, the lysates were incubated with Flag-agarose. Immunocomplexes were washed with lysis buffer three times and eluted with 2 × SDS sample buffer. Antibodies used for this study were Acetylated lysine (Cell Signaling Technology, Danvers, MA, USA (CST), 9814), Trx1 (CST, 2429), mTOR (CST, 2797), Phospho-mTOR (Ser2481) (CST, 2774), p70 S6 Kinase (CST, 9202), Phospho-p70 S6 Kinase (CST, 9205), 4EBP1 (CST, 9644), Phospho-4EBP1 (CST, 2855), AMPKα2 (CST, 2757), Phospho-AMPKα (CST, 2535), ACC (CST, 3662), Phospho-ACC (CST, 3661), Gapdh (CST, 5174), Prdx1 (Abcam, Cambridge, UK, ab41906), Sirt1 (EMD Millipore, Burlington, MA, USA, 07-131), Nampt (Abcam, ab58640), Namnat1 (Bethyl, Montgomery, TX, USA, A304-317A), Tubulin (Sigma, T6199), α-cardiac actin (Novus, Saint Louis, MO, USA, NBP2-61474), and HDAC1 (BioVision, Milpitas, CA, USA, 3601-30T). The signal intensity of the Western blot signals was quantified using the ImageJ program (Bethesda, MD, USA). The signal intensity of the non-phosphorylated proteins was normalized by a loading control (Tubulin). The signal intensity of the phosphorylated or acetylated proteins was normalized by that of the relevant total protein. Normalized signal intensity relative to the control was used for statistical analyses of one or more Western blot membranes.

### 2.3. Adenovirus Vectors

Adenovirus vectors expressing Flag- and HA-Tagged Trx1 (Flag-Trx1-HA) and shTrx1 were generated with the AdMax system as described previously [7]. Ad-shHDAC1 was generated using 19 nucleotides corresponding to the mouse HDAC1 N-terminal region.

### 2.4. Ketogenic Diet

C57BL/6 mice (11-week-old) were fed a control (D10070802) or ketogenic diet (D10070801) for 5 days. Diets were purchased from Research Diets, Inc (New Brunswick, NJ, USA). The control diet consisted of 10% protein, 80% carbohydrates and 10% fat, whereas ketogenic diet consisted of 10% protein and 90% fat [17]. All procedures involving animals were performed in accordance with protocols approved by Rutgers Biomedical and Health Sciences (Protocol Number: 999900700).

### 2.5. Statistical Methods

Statistical comparisons were made using Student’s *t* test for pairwise and one-way ANOVA for multiple comparisons. *p* < 0.05 was defined as statistically significant and is indicated by a filled asterisk. All error bars represent S.E.M.

## 3. Results

### 3.1. βHB Upregulates Trx1

To examine the effect of βHB upon Trx1 expression, primary cultured cardiomyocytes were treated with 0.1 to 10 mM βHB for 16 h. The circulating level of βHB is 0.4–0.5 mM in non-diabetic patients at baseline [18]. This is significantly higher in diabetic patients at baseline or in response to fasting in non-diabetic patients. We have previously shown that 1 mM βHB affects signaling in cardiomyocytes [10]. Therefore, we chose dosages of βHB ranging from 0.1 to 10 mM. βHB upregulated Trx1 in a dose-dependent manner from 0.1 mM to 3 mM, whereas 10 mM βHB induced Trx1 less effectively than other dosages (Figure 1A). βHB also upregulated Prdx1, a Trx1 substrate, at 0.1 and 0.3 mM. In contrast, the levels of other factors previously reported to act protectively against diabetic cardiomyopathy, including Sirt1, Nampt, and Namnat1, were not significantly changed by the βHB treatment. βHB (1 mM) also upregulated Trx1 in a time-dependent manner (Figure 1B). Significant upregulation of Trx1 was observed as early as 20 min after βHB treatment, reached a plateau at 1 h, and was sustained for 24 h, the longest time point evaluated. Thus, βHB upregulates Trx1 in cardiomyocytes in a dose- and time-dependent manner. To examine whether βHB also upregulates Trx1 in cardiac fibroblasts, cells were treated with 3 mM βHB for 16 h. βHB-induced Trx1 upregulation was also observed in cardiac fibroblasts (Figure 1C). Ketogenic diet consumption for 5 days also upregulated Trx1 in mouse hearts in vivo (Figure 1D). Taken together, these results suggest that βHB, a ketone body, upregulates Trx1 in the heart in a cell autonomous manner.

### 3.2. βHB Confers Resistance against Oxidative Stress through a Trx1-Dependent Manner

To examine whether βHB induces resistance against oxidative stress in a Trx1-dependent manner, cardiomyocytes were treated with H_2_O_2_ in the presence of βHB with or without short-hairpin RNA (shRNA)-mediated downregulation of Trx1. We confirmed that βHB-induced Trx1 upregulation was abolished in the presence of adenovirus harboring Trx1 shRNA (shTrx1) but did not control shRNA (Figure 2A). βHB attenuated H_2_O_2_-induced cardiomyocyte death, as evaluated with trypan blue staining, but the effect was abolished in the presence of Trx1 knockdown (Figure 2B). Compared to 1 mM βHB, 10 mM βHB did not significantly attenuate H_2_O_2_-induced cardiomyocyte death (Figure 2C). This is consistent with the fact that 10 mM βHB no longer significantly upregulates Trx1 compared to 0.3–3 mM βHB (Figure 1A). These results suggest that βHB protects cardiomyocytes against oxidative stress through endogenous Trx1.

### 3.3. βHB Prevents H_2_O_2_-Induced mTOR Inhibition in a Trx1-Dependent Manner

To investigate whether βHB potentiates Trx1 functions, the effect of βHB on mTOR, a known Trx1 substrate [6], was examined. As shown previously, following H_2_O_2_ treatment, the mTOR in cardiomyocytes exhibited a band shift to higher molecular weights in SDS-PAGE analyses under non-reducing conditions, whereas the band shift was abolished under reducing conditions in the presence of 2-mercaptoethanol (2ME) (Figure 3A) [6]. These results suggest that mTOR forms intermolecular disulfide bonds upon H_2_O_2_ treatment. The band shift is promoted by Trx1 knockdown, indicating that mTOR is a Trx1 substrate [6]. Although βHB prevented the H_2_O_2_-induced mTOR band shift, this effect was abolished in the presence of Trx1 knockdown (Figure 3B). βHB also normalized H_2_O_2_-induced decreases in the phosphorylation of mTOR and its substrates, including S6K and 4EBP1. Again, the effect of βHB was abolished in the presence of Trx1 knockdown (Figure 3C). These results suggest that βHB prevents the oxidation and inactivation of mTOR in response to H_2_O_2_ in a Trx1-dependent manner. These results are consistent with the notion that βHB potentiates Trx1 function.

### 3.4. βHB Prevents H_2_O_2_-Induced AMPK Inhibition in a Trx1-Dependent Manner

To further confirm that βHB potentiates Trx1 function, the effect of βHB on AMPK, another Trx1 substrate [7], was examined. As shown previously, AMPKα exhibited band shifts to higher molecular weights in SDS-PAGE analyses under non-reducing conditions following H_2_O_2_ treatment. The band shifts were abolished when the SDS-PAGE analyses were conducted in the presence of 2ME (Figure 4A) [7]. These results suggest that AMPKα forms intermolecular disulfide bonds upon H_2_O_2_ treatment. βHB prevented the H_2_O_2_-induced AMPKα band shifts, an effect that was abolished in the presence of Trx1 knockdown (Figure 4B). βHB also normalized H_2_O_2_-induced decreases in the phosphorylation of ACC, an AMPK substrate [19], but the effect was abolished in the presence of Trx1 knockdown (Figure 4C). These results suggest that βHB maintains AMPK function in a Trx1-dependent manner.

### 3.5. βHB May Upregulate Trx1 through HDAC Inhibition

We then investigated the mechanism through which βHB upregulates Trx1 in cardiomyocytes. Since βHB inhibits the HDAC activity of class I HDACs [14], βHB may upregulate Trx1 through HDAC inhibition. Consistent with our hypothesis, βHB promoted protein acetylation in a dose-dependent manner. βHB increased HDAC1 protein, which likely represents compensatory feedback regulation. The dose-dependency of the effect of βHB on the induction of protein acetylation (Figure 5A) was similar to that observed in the upregulation of Trx1. In contrast to βHB, 30 and 100 µM H_2_O_2_ did not significantly upregulate HDAC1, whereas 300 µM H_2_O_2_ downregulated HDAC1 (Figure 5B). To investigate whether HDAC1 inhibition upregulates Trx1, HDAC1 was knocked down with HDAC1 shRNA in the cardiomyocytes. The knockdown of HDAC1 upregulated Trx1 in a dose-dependent manner (Figure 5C). To test whether βHB induces Trx1 acetylation, Flag- and HA-tagged Trx1 was overexpressed in cardiomyocytes using an adenovirus vector. Trx1 was immunoprecipitated with anti-Flag antibody and then subjected to immunoblot analyses with anti-acetylated lysine antibody and anti-HA antibody. βHB significantly increased acetylated Trx1/total Trx1, indicating that βHB induces Trx1 acetylation (Figure 5D). Since Trx1 upregulation is observed shortly after βHB treatment, βHB may upregulate Trx1 with a posttranslational mechanism. To test this possibility, cardiomyocytes were treated with βHB and cycloheximide (CHX), an inhibitor of protein translation. Degradation of the Trx1 protein in cardiomyocytes in the presence of CHX was significantly retarded in the presence of βHB (Figure 5E), suggesting that βHB may upregulate Trx1 by inhibiting protein degradation.

## 4. Discussion

### 4.1. βHB Upregulates Trx1

Ketone bodies have emerged as beneficial nutrients in cardiac pathophysiology. A modest increase in circulating ketone bodies resulting from ketogenic diet consumption mitigates oxidative stress in the heart, as evidenced by decreases in oxidized glutathione, 3-nitrotyrosine, and malondialdehyde levels [20]. The current study shows that βHB at 0.3–3 mM upregulates Trx1 and confers resistance against oxidative stress through Trx1 in cultured cardiomyocytes. βHB potentiated Trx1 function in the presence of oxidative stress, as evidenced by the rescue of oxidation-induced inactivation of mTOR and AMPK, known direct substrates of Trx1. Although the extent to which how Trx1 contributes to biological functions of mTOR and AMPK such as cell growth and fatty acid oxidation needs to be further investigated. Our previous works have shown that Trx1 confers resistance against oxidative stress, partly through mTOR and AMPK [6,7]. Thus, we propose that βHB confers stress resistance to cardiomyocytes, partly through upregulation of the Trx1 system (Figure 6).

### 4.2. Ketone Bodies in mTOR Regulation

We showed that βHB normalizes the H_2_O_2_-induced oxidative inhibition of mTOR via Trx1. mTOR is inhibited by the direct oxidation at Cys1483, whereas Trx1 normalizes mTOR activity via the reduction of Cys1483 [6]. The Trx1-mediated normalization of mTOR may alleviate oxidative stress-induced downregulation of mitochondrial genes [6], which may protect the heart under diabetic conditions. It should be noted that the overall effect of ketone bodies upon mTOR appears to be context dependent. Since mTOR is regulated by multiple mechanisms, βHB may affect the overall activity of mTOR through multiple mechanisms, both positively and negatively. For example, βHB inhibits mTOR activation and cardiomyocyte hypertrophy induced by phenylephrine, an agonist of the α1-adrenergic receptor [10]. Phenylephrine-induced mTOR activation is mediated by Erk, and phenylephrine-induced Erk activation is sensitive to antioxidant treatment [21,22]. Thus, it is possible that βHB inhibits phenylephrine-induced mTOR activation through the suppression of Erk, which is facilitated by the maintenance of the redox environment by the Trx1 system.

### 4.3. Ketone Body in Diabetic Cardiomyopathy

Trx1 tended to be upregulated at 0.1 mM βHB, whereas it was significantly upregulated at 0.3 mM βHB in cardiomyocytes (Figure 1A). The plasma level of βHB is 0.4–0.5 mM in non-diabetic patients at baseline. It is significantly elevated in diabetic patients or in response to fasting in non-diabetic patients. Thus, it is likely that βHB could affect the level of Trx1 in cardiomyocytes under pathophysiologically relevant conditions. Despite the generally salutary effect of ketone bodies in the heart, whether ketone bodies are also beneficial in diabetic cardiomyopathy is not fully understood. Further investigation is needed to demonstrate that modest elevation of the plasma level of βHB protects the heart through the upregulation of Trx1 in vivo. It should be noted, however, that uncontrolled ketone body synthesis during diabetes results in ketoacidosis, where βHB reaches over 3 mM in the blood plasma [23]. We show here that βHB confers resistance against oxidative stress in cardiomyocytes through the upregulation of Trx1. It should be noted, however, that a high dose of βHB (10 mM) was less effective in inducing Trx1 and Prdx1 as well as overall protein acetylation levels than lower doses of βHB (Figure 1A and Figure 5A). Indeed, a high dose of βHB (10 mM) did not significantly confer resistance against oxidative stress (Figure 2C). Taken together, the data indicate that while ketone bodies at physiological concentrations and at the slightly elevated (1–3 mM) concentrations seen in diabetic patients mitigate oxidative stress, they attenuate the ability to promote Trx1 at the high concentrations (>3 mM) seen during ketoacidosis. Although ketone bodies may act as a thrifty substrate and may improve energetics [13], our results suggest that βHB also has an additional effect, namely, to stimulate Trx1, thereby conferring stress resistance to the heart.

### 4.4. Ketone Bodies, HDAC Inhibtioin and Trx1 as a Longevity Factor

Calorie restriction and fasting promote ketone body synthesis, which potentially mediates the salutary effects of calorie restriction, including lifespan extension. In rats, calorie restriction induces a modest elevation of βHB, approximately 0.7 mM, in blood plasma [24]. βHB is a natural inhibitor of class I HDACs such as HDAC1 and HDAC2 [14], and the inhibition of class I HDACs enhances cardiac function and longevity in Drosophila melanogaster [25]. We show here that both βHB and HDAC1 inhibition upregulate Trx1. Interestingly, both a ketogenic diet and systemic Trx1 overexpression have been shown to reduce midlife mortality without affecting maximum lifespan in mice [26,27]. Thus, it will be interesting to investigate whether Trx1 mediates the salutary actions of calorie restriction, ketogenic diet, and HDAC inhibition.

### 4.5. Mechanism Responsible for Trx1 Upregulation

Our results show that βHB induces Trx1 acetylation and inhibits Trx1 degradation. Besides inhibiting HDACs, βHB may induce protein acetylation via an increase in acetyl-CoA [9]. It has been shown previously that Trx1 is acetylated at several lysine residues, including Lys94 [28], and that Trx1 is ubiquitinated at Lys94. These results suggest that Trx1 may be degraded through the ubiquitin–proteasome pathway [29]. Since Lys94 can be either acetylated or ubiquitinated, βHB-induced acetylation at Lys94 may compete with other mechanisms promoting ubiquitination at Lys94, thereby controlling the stability of Trx1. Further investigations are necessary to test this possibility.

### 4.6. Experimental Limitation

We assessed the activity of mTOR and AMPK to evaluate the oxidoreductase activity of Trx1 in this study. It should be noted, however, that mTOR and AMPK may be only one of the many effects of βHB, and we have not evaluated how βHB affects cellular function through mTOR and AMPK in this study. Thus, further investigation is needed to elucidate how the βHB–Trx1 axis protects the heart against diabetic cardiomyopathy.

## 5. Conclusions

βHB, a major ketone body in mammals, confers resistance against oxidative stress to cardiomyocytes through Trx1 induction. βHB upregulates Trx1, possibly through class I HDAC inhibition and through the inhibition of Trx1 degradation.

## Figures and Tables

**Figure 1 antioxidants-10-01153-f001:**
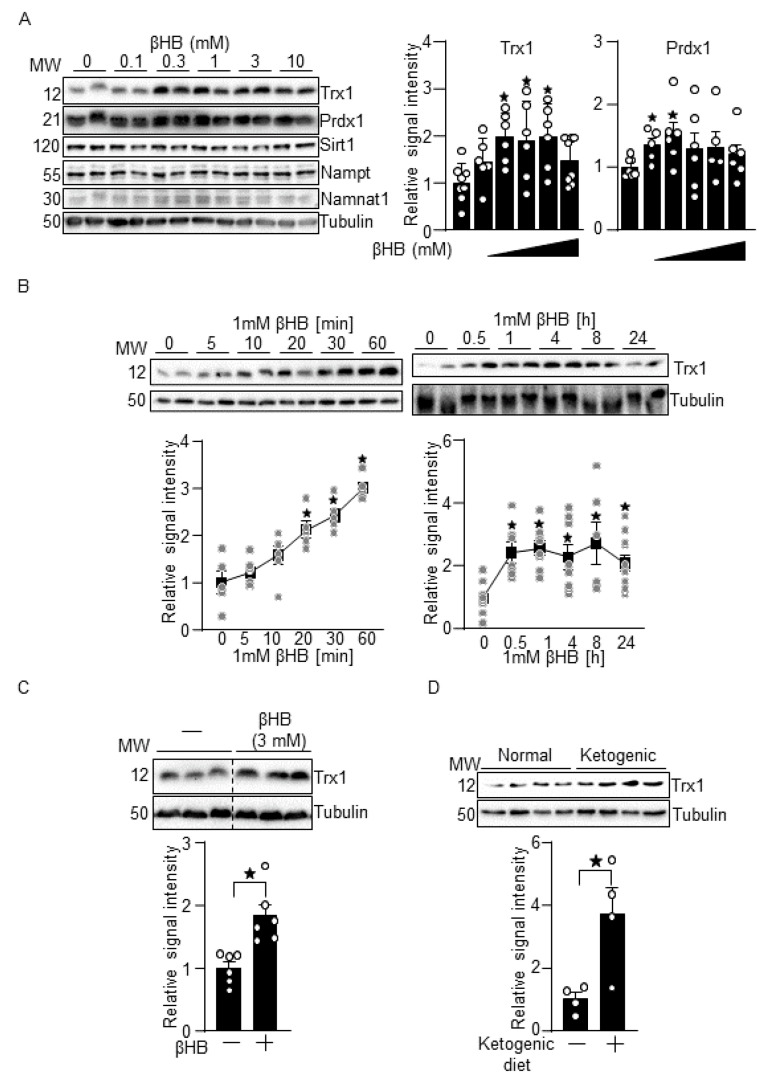
βHB upregulates Trx1. (**A**) Dosage effect of βHB on Trx1 expression. Cardiomyocytes were treated with the indicated concentrations of βHB for 16 h. *n* = 6–8 (Trx1) and 5–7 (Prdx1). (**B**) Time-dependent effect of 1 mM βHB on Trx1 in cardiomyocytes. *n* = 5–7. (**C**) βHB upregulates Trx1 in cardiac fibroblasts. Cardiac fibroblasts were treated with 3 mM βHB for 16 h. *n* = 6. (**D**) Ketogenic diet consumption upregulates Trx1 in the heart in vivo. *n* = 4. Western blot analyses with indicated antibodies were performed. * *p* < 0.05 (**A**–**D**).

**Figure 2 antioxidants-10-01153-f002:**
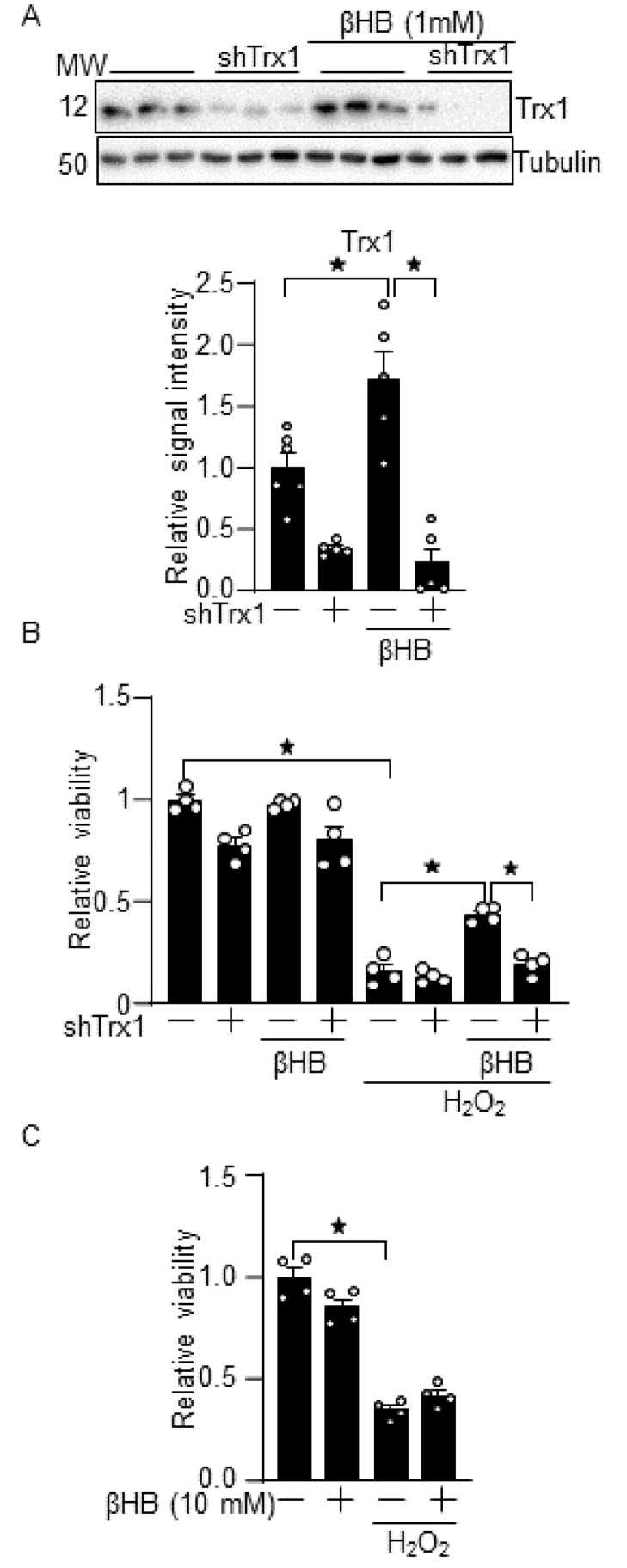
βHB potentiates the antioxidant defense in a Trx1-dependent manner. (**A**) Trx1 knockdown inhibits βHB-induced Trx1 upregulation. Cardiomyocytes were treated with Trx1 short hairpin RNA (shTrx1) adenovirus vector for 3 days and with 1 mM βHB for 16 h. The expression of Trx1 was examined using Western blot analysis. *n* = 5–6. (**B**) βHB inhibits H_2_O_2_-induced cardiomyocyte death in a Trx1-dependent manner. (**C**) A high dose of βHB (10 mM) did not significantly inhibit H_2_O_2_-induced cardiomyocyte death. (**B**,**C**) Cardiomyocytes were treated with 1 or 10 mM βHB for 16 h and were then treated with 100 µM H_2_O_2_ for 6 h. The cardiomyocyte viability was examined using trypan blue dye exclusion. *n* = 4. * *p* < 0.05 (**A**–**C**).

**Figure 3 antioxidants-10-01153-f003:**
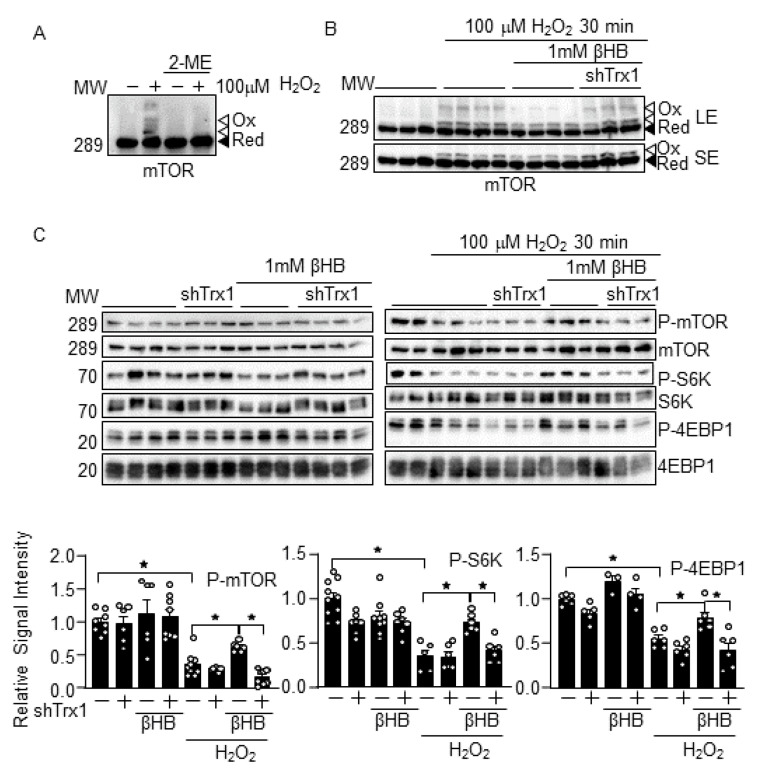
βHB potentiates mTOR function in a Trx1-dependent manner. (**A**) mTOR exhibits intermolecular disulfide bond formation upon oxidation. Cardiomyocytes were treated with 100 µM H_2_O_2_ for 30 min. Western blot analyses were performed following SDS-PAGE under non-reducing (without 2-ME) and reducing (with 2-ME) conditions. (**B**) βHB inhibits H_2_O_2_-induced mTOR oxidation in a Trx1-dependent manner. Cardiomyocytes were treated with 1 mM βHB for 4 h and were then treated with 100 µM H_2_O_2_ for 30 min. LE: long exposure, SE: short exposure. (**C**) βHB normalizes H_2_O_2_-induced mTOR inhibition in a Trx1-dependent manner. Western blot analyses were performed with indicated antibodies. *n* = 6–9 (p-mTOR), 3–6 (P-S6K), and 3–6 (P-4EBP1). * *p* < 0.05 (**C**).

**Figure 4 antioxidants-10-01153-f004:**
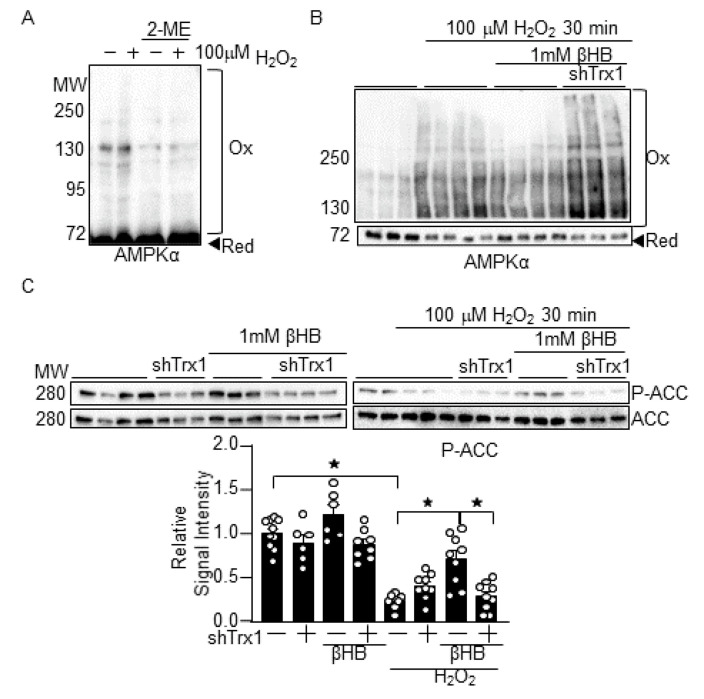
βHB potentiates AMPK function in a Trx1-dependent manner. (**A**) AMPKα exhibits intermolecular disulfide bond formation upon oxidation. Cardiomyocytes were treated with 100 µM H_2_O_2_ for 30 min. Western blot analyses were performed following SDS-PAGE under non-reducing (without 2-ME) and reducing (with 2-ME) conditions. (**B**) βHB inhibits H_2_O_2_-induced AMPKα oxidation in a Trx1-dependent manner. (**C**) βHB normalizes H_2_O_2_-induced AMPK inhibition in a Trx1-dependent manner. *n* = 6–10. Western blot analyses were performed with indicated antibodies. * *p* < 0.05 (**C**).

**Figure 5 antioxidants-10-01153-f005:**
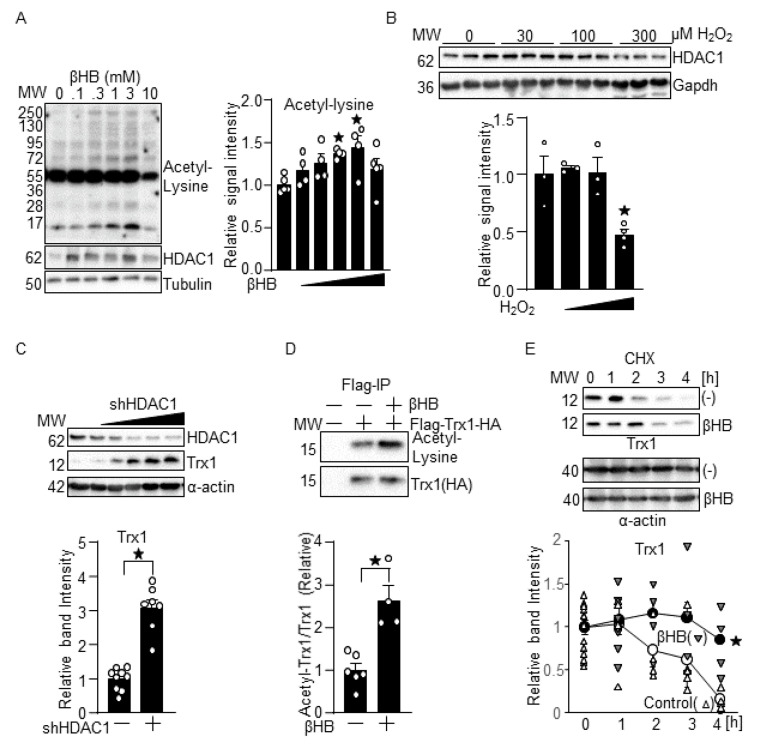
βHB may upregulate Trx1 through HDAC inhibition. (**A**) Dosage effects of βHB on protein acetylation and HDAC1 expression. Cardiomyocytes were treated with the indicated concentrations of βHB for 16 h. *n* = 4–5. (**B**) Dosage effects of H_2_O_2_ upon HDAC1 expression. Cardiomyocytes were treated with the indicated concentrations of H_2_O_2_ for 16 h. *n* = 3–4. (**C**) Knockdown of HDAC1 upregulates Trx1. (Left) Cardiomyocytes were transduced with 0, 0.3, 1, 3, 10, or 30 MOI of shHDAC1 for 3 days. (Right) Trx1 induction by shHDAC1 (3 MOI). *n* = 7–9. (**D**) βHB induces Trx1 acetylation. Flag- and HA-tagged Trx1 was expressed with adenovirus vector and immunoprecipitated with anti-Flag antibody. *n* = 4–6. (**E**) βHB inhibits Trx1 degradation. Cardiomyocytes were treated with 100 µM cycloheximide (CHX) for 1 h followed by 1 mM βHB for 1 to 4 h. *n* = 4–5. (**A**–**E**) Western blot analyses were performed with indicated antibodies. * *p* < 0.05 (**A**–**E**).

**Figure 6 antioxidants-10-01153-f006:**
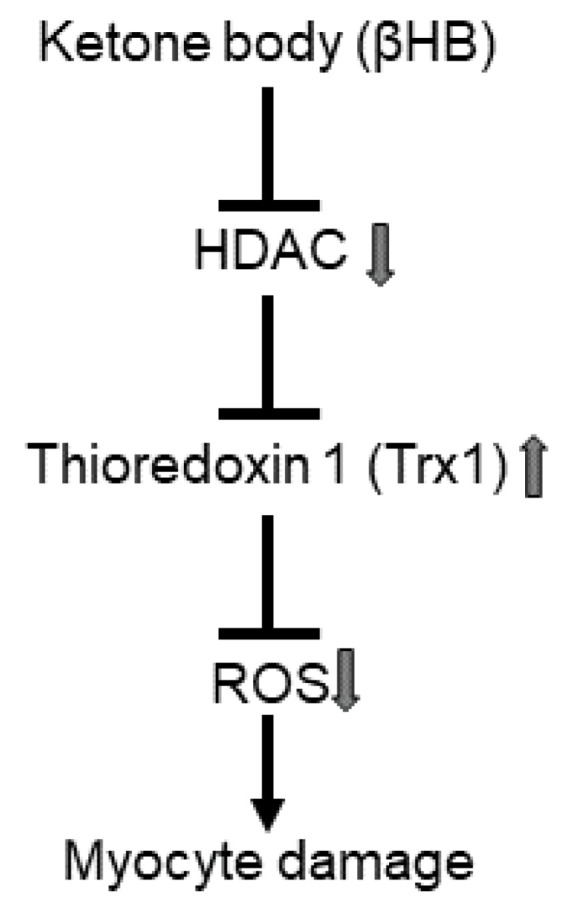
Schematic representation of ketone body-induced antioxidant defense. Ketone body upregulates Trx1, possibly through HDAC inhibition, thereby potentiating the antioxidant defense.

## Data Availability

The authors declare that all supporting data are available within the article. In addition, any raw data that support the findings of this study are available from the corresponding author upon reasonable request.

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
