# Peer review of "β-Hydroxybutyrate, a Ketone Body, Potentiates the Antioxidant Defense via Thioredoxin 1 Upregulation in Cardiomyocytes"

_antioxidants, 2021, doi:10.3390/antiox10071153_

Round 1

Reviewer 1 Report

In the current manuscript, Oka et al. investigate the role of ketone bodies on the regulation of Trx1 leading to antioxidative protection of cardiomyocytes. The subject is of great interest and provide a very novel regulatory mechanism involving ketone bodies. The results are well presented and support the conclusions. The main limitations of the current study lie on the in vitro exploration and thus a lack of in vivo evidence of the beneficial effect proposed.  In addition, the final proposed mechanism involving acetylation of Trx1 remains hypothetical although strongly expected.

Nevertheless, I would suggest some additions to improve the significance and understanding of the manuscript.

  1. Authors should discuss the dose of βHB chosen. In addition, the effective dose of 0.1mM should be discuss based on In vivo levels of βHB measured either in normal conditions and during ketogenic diets or a fast. Since the authors discuss a putative high vs low dose effect of β
  2. Following the previous question, what would be the effect of a high dose of βHB on the measured parameters. I believe that adding results of high doses would support the importance of the level of βHB and its putative dual effects. Indeed, in vivo experiments would need such preliminary results to avoid damaging effects.
  3. Direct acetylation of Trx1 could have been assessed using an Immunoprecipitation method.
  4. The main weakness of the study comes from the lack of in vivo evidence of the proposed mechanism. At least, a ketogenic diet followed by the measurement of trx1 levels as well as acetylation and oxidation tests showed in the current paper could have been done. Indeed, βHB seems to have an effect in cardiomyocytes even in normal conditions. Such addition would strongly improve the significance of the work presented here.

Author Response

Reviewer 1

  1. Authors should discuss the dose of βHB chosen. In addition, the effective dose of 0.1mM should be discuss based on in vivo levels of βHB measured either in normal conditions and during ketogenic diets or a fast. Since the authors discuss a putative high vs low dose effect of βHB.

The reviewer raised an important issue. The circulating level of bHB is 0.4-0.5 mM in non-diabetic patients at baseline. It is significantly higher in diabetic patients at baseline or in response to fasting in non-diabetic patients. We have shown previously that 1 mM βHB affects signaling in cardiomyocytes (Nakamura et al, Cardiovascular Research 2020). Therefore, we chose dosages of βHB ranging from 0.1 to 10 mM.

In addition, we discussed the effective dose of 0.1 mM βHB as follows (lines 292-296). “Trx1 tended to be upregulated at 0.1 mM βHB whereas it was significantly upregulated at 0.3 mM βHB in cardiomyocytes (Fig. 1A). The plasma level of βHB is 0.4-0.5 mM in non-diabetic patients at baseline. It is significantly elevated in diabetic patients or in response to fasting in non-diabetic patients. Thus, it is likely that βHB could affect the level of Trx1 in cardiomyocytes under pathophysiologically relevant conditions.”

  1. Following the previous question, what would be the effect of a high dose of βHB on the measured parameters. I believe that adding results of high doses would support the importance of the level of βHB and its putative dual effects. Indeed, in vivo experiments would need such preliminary results to avoid damaging effects.

We agree with the reviewer. bHB-induced upregulation of Trx1 was blunted at 10 mM bHB after reaching a peak at 3 mM bHB. We examined the effect of 10 mM bHB in H2O2-induced cell death. The high dose of βHB did not significantly inhibit cell death.  This is consistent with the fact that 10 mM βHB no longer significantly upregulates Trx1 compared to 0.3-3 mM βHB (Fig. 1A).

  1. Direct acetylation of Trx1 could have been assessed using an Immunoprecipitation method.

We have originally assessed Trx1 acetylation with immunoblots following immunoprecipitation with anti-Flag antibody (Fig. 5C).

  1. The main weakness of the study comes from the lack of in vivoevidence of the proposed mechanism. At least, a ketogenic diet followed by the measurement of trx1 levels as well as acetylation and oxidation tests showed in the current paper could have been done. Indeed, βHB seems to have an effect in cardiomyocytes even in normal conditions. Such addition would strongly improve the significance of the work presented here.

We agree with the reviewer. We conducted an additional experiment and evaluated the extent to which a ketogenic diet upregulates Trx1 in the mouse heart in vivo. Consumption of a ketogenic diet for 5 days significantly upregulated Trx1 in the heart (Fig. 1D).

Reviewer 2 Report

This is an interesting work in which the authors study the effects of β-hydroxybutyrate (βHB) on the activation of Thioredoxin 1 (Trx1) in primary cultured cardiomyocytes. To do this, they study the oxidation levels of different Trx1 targets such as mTOR or AMPKα. In addition, the authors try to explain the possible mechanism of regulation of Trx1 through the inhibition of HDAC1 and its consequences on acetylation. Overall, the experiments were well-designed and the results are interesting that contribute significantly to our understanding of the action of βHB in protection of cardiomyocytes.

However, I have some comments that need to be addressed:

  • First, the manuscript tittle. Talk about ketone bodies when only βHB has been tested, and antioxidant defense, when only TRX1 up-regulation has been explored. I suggest that it should be reviewed.
  • Due to the antioxidant properties of βHB, it would be advisable to show some determination of oxidation parameters, for example the GSSG / GSH ratio, malondyladehyde or a protein carbonylation pattern.
  • In Figures 3B and 4B, it would be interesting to see what happens when shTrx1 is used and incubated with hydrogen peroxide without the presence of βHB. Would mTOR oxidation exist?

I comment on it because that condition appears in Figures 3C and 4C.

  • Regarding the histograms in Figures 3C and 4C, what have you done to include in the same graph the relative intensities of different WB membranes?
  • In section 3.3 the authors comment on the recovery of mTOR function, due to the antioxidant function of Trx1, through the phosphorylation of its targets. These proteins are involved in cell growth and proliferation. It would be interesting to see in this work the impact on these cellular processes and thus verify that the functionality of the pathway is recovered.
  • According to what was commented in section 3.3, the same could be done in 3.4. The phosphorylation and therefore inhibition of ACC, what impact does it have?
  • In Figure 5D, Trx1 levels are shown after incubation with CHX. Are these membranes from the same gel?

In addition, it would be convenient to review the statistics because at the point where the greatest visual change is observed (2h), there is no significant difference. On the other hand, it exists at 4h, where it seems like there are no changes.

Author Response

Reviewer 2

  1. First, the manuscript tittle. Talk about ketone bodies when only βHB has been tested, and antioxidant defense, when only TRX1 up-regulation has been explored. I suggest that it should be reviewed.

We have revised the title as follows. Thank you very much.

β-hydroxybutyrate, a ketone body, potentiates the antioxidant defense via Thioredoxin 1 upregulation in cardiomyocytes.

  1. Due to the antioxidant properties of βHB, it would be advisable to show some determination of oxidation parameters, for example the GSSG/GSH ratio, malondyladehyde or a protein carbonylation pattern.

We agree with the reviewer. It has been shown that a ketogenic diet mitigates oxidative stress in the heart, as evidenced by decreases in oxidized glutathione and malondialdehyde. We cited this literature in the Discussion (4.1). 

  1. In Figures 3B and 4B, it would be interesting to see what happens when shTrx1 is used and incubated with hydrogen peroxide without the presence of βHB. Would mTOR oxidation exist?

We have shown previously that H2O2-induced mTOR oxidation (a band shift) is enhanced by shTrx1. We cite our paper in the Results (3.3).

  1. Regarding the histograms in Figures 3C and 4C, what have you done to include in the same graph the relative intensities of different WB membranes?

Signal intensity relative to control (left 3 lanes for the left panel and left 2 lanes for the right panel) was used for statistical analyses of either the same or different Western blot membranes. We have described this in Materials and Methods (lines 110-111).

  1. In section 3.3 the authors comment on the recovery of mTOR function, due to the antioxidant function of Trx1, through the phosphorylation of its targets. These proteins are involved in cell growth and proliferation. It would be interesting to see in this work the impact on these cellular processes and thus verify that the functionality of the pathway is recovered.

How the rescue of mTOR activity by bHB at baseline affects growth and proliferation is interesting. In this study, we used phosphorylation of mTOR and its substrates as indexes of the Trx1 activity. However, how bHB affects the overall function of mTOR and how Trx1 contributes to it are complex. We have discussed this issue in the Discussion (4.2).

  1. According to what was commented in section 3.3, the same could be done in 3.4. The phosphorylation and therefore inhibition of ACC, what impact does it have?

How the rescue of AMPK activity by bHB at baseline affects growth and proliferation is certainly interesting. In this study, we used phosphorylation of AMPK and ACC as indexes of the Trx1 activity. However, addressing how bHB affects the overall function of fatty acid oxidation and how Trx1 contributes to it requires extensive experimentation. We have discussed this issue in the Discussion (4.1).

  1. In Figure 5D, Trx1 levels are shown after incubation with CHX. Are these membranes from the same gel?

In addition, it would be convenient to review the statistics because at the point where the greatest visual change is observed (2h), there is no significant difference. On the other hand, it exists at 4h, where it seems like there are no changes.

            These images were derived from the same gel/membrane as shown below.

Although the Western blot images shown in Figure 5D are the most representative, we experienced a certain degree of variation. We now show individual data points in a dot plot format.

Reviewer 3 Report

This study aims to determine whether ketone body-induced thioredoxin 1 (Trx1) mediates the beneficial roles to oxidative stress. Although the antioxidant and anti-inflammatory capacities of ketone diet have been suggested, it remains to be confirmed in cardiac cells, and to be fully elucidated the underlying molecular mechanisms. Authors demonstrated that ketone body acts as a protective agent against oxidative stress to the cardiomyocytes and identified that ketone body-induced downregulation of HDAC1 may stabilize Trx1 expression. The experimental design and data are well organized. Overall findings are notable and commendable. This would be beneficial for developing new therapeutic approaches based on the efficacy of the ketone diet. 

One concern I have is the validity of this study model. Due to the differences in metabolic preferences between neonate and adult myocardium, there is a possibility that the findings in neonatal primary culture cells may not recapitulate the clinical settings. Moreover, given that this special issue “antioxidants in diabetes” focuses on diabetic stress or injury, it may be necessary to consider the direct pathogenesis of diabetic cardiomyopathy, such as hyperglycemia and dyslipidemia, in addition to H2O2 stress. It would be desirable to confirm the fundamental observation that ketone bodies prevent cardiac injury in response to H2O2 and at least a diabetic stress in an adult cardiomyocyte or in vivo model.

Specific concerns:
1. Some key findings should be confirmed in adult cardiomyocytes or in vivo to see if it’s reproducible. 
2. This manuscript is submitted for Special Issue "Antioxidants in Diabetes". An addressing test will be appreciated to confirm how Trx1 behaves in the pathogenesis of obesity and diabetes mellitus. Is the Trx1 expression increased by βHB treatment in high glucose or high fat stress? And if so, will that mitigate the toxicity?

Minor concerns: 
3. The title sounds too general. It could be more specific about the effects in cardiomyocyte. 
4. Some of the immunoblots shows a large variation between the duplicates; Fig1A Prdx1, 1B right panel Tubulin, would need to be replaced. 
5. Please add the method of cell viability assay.
6. It is uncertain which group the two leftmost lanes in the right panels of Fig3C and Fig4C belong to. 
7. In Fig5D, what was the loading control and how was the measurements normalized?
8. In which figure/experiment were the heart homogenates (described in the methods 2.2) used?
9. The half-life of β-hydroxybutyrate (βHB) is relatively short (1-3hrs, PMID:22561291). The time course of Trx1 expression seems to reach almost plateau at 1h after the start of treatment, and the tendency seems to be slight downward between 8h and 24h. Please explain the rationale for choosing 16hrs treatment in the following experiments.
10. Fig5. How was the change in expression of HDAC1 during H2O2 treatment? 
11. Fig6. Please reconsider the structure of diagram. It is a little confusing whether mTOR and AMPK mediates the antioxidative capacity of Trx1.

Round 2

Reviewer 1 Report

In the newly submitted manuscript, the authors have addressed all my comments.

No further comments

Reviewer 2 Report

First of all, I would like to indicate that the authors have responded correctly to the suggestions indicated. Although I would have liked to see the consequences of mTOR and AMPK activation due to Trx1 upregulation, I consider that the manuscript presents sufficiently strong conclusions to be accepted.

Reviewer 3 Report

The revised manuscript satisfactory addressed my concerns.